# A Study on the Subjectivity of Parents Regarding “0th-Period Physical Education Class” of Middle Schools in Korea Using Q-Methodology

**DOI:** 10.3390/ijerph19137760

**Published:** 2022-06-24

**Authors:** Wonseok Choi, Wonjae Jeon

**Affiliations:** 1Department of Physical Education, Keimyung University, Daegu 42601, Korea; wschoi@kmu.ac.kr; 2Department of Physical Education, Korea National University of Education, Cheongju-si 28173, Korea

**Keywords:** 0th-period physical education class, adolescent, parents, Q-methodology, subjectivity

## Abstract

The current study examined parents’ subjective perception types and characteristics regarding the 0th-Period Physical Education Class of Middle School in Korea. The Q-methodology was applied, and the final 25 Q-Samples were selected through the composition of the 42 Q-population. Among Korean parents, 20 students who participated in “Physical Education Activities in Class 0” for more than one year were selected as P-Sample. Q-sorting was performed by the P-Sample. Data collected by Q-sorting were analyzed using the PQ method program version 2.35, with centroid factor analysis and varimax rotation. The finding pointed to four types, with a total explanatory variance of 63%. Type 1 (N = 7), and was named “urgent legal and institutional settlement of 0th-period physical education”. Type 2 (N = 4) has the theme of “beach-head for a vibrant school life”. Type 3 (N = 4) was named “enhancement of academic capability”. Type 4 (N = 4) was described as “strengthening physical and mental health”. Moreover, the consensus statements between each type were investigated in Q1 and Q2. These findings highlight the importance of the “0th-period physical education class” so the program could be expanded and institutionalized in Korea.

## 1. Introduction

Different cardiovascular exercises that can be practiced in everyday life, such as swimming, walking, running, cycling, and mountain climbing, improve oxygen transfer to each body cell [1]. In addition, it positively affects neurological and cognitive function in all ages, from children to the elderly [2]. Since regular exercise has been steadily reported to improve brain function in various age groups, adolescence, the most active period for nerve production and cell connection, is a critical moment to enhance cognitive function by participating in the exercise [3].

Physical activity during adolescence positively affects the brain’s readiness to accept new information, improving learning memory [4] and optimizing the mental environment to increase alertness and concentration [5]. Marsh and Kleitman [6] noted that healthy children generally have better learning skills than their counterparts. Furthermore, children who practiced regular exercise also had higher memory skills and academic achievement [7]. Hence, it is well documented that adolescent exercise improves the brain’s structure by supplying blood to the brain and further improves academic ability while functioning to make the brain’s state optimal [8].

One interesting topic among the various studies of adolescent exercise participation is related to morning exercise. For example, a recent study revealed that routine morning exercise improved the health and weight management of the overweight population [9]. Moreover, McGowan et al. [10] suggested that completion of a morning swimming session with resistance exercise can substantially enhance sprint-swimming performance completed later the same day. In this respect, morning exercise can draw academic significance in studies targeting adolescents. For instance, consistent morning exercise participation, such as swimming in adolescents, positively affects the sleep–wake cycle [11]. In addition, Babadi et al. [12] observed that adolescents who engaged in morning exercise at school had a lower prevalence of hypertension. At the same time, the other group who spent more time on sedentary activities were at higher risk for hypertension.

A project attempted to prove the effectiveness of such early morning exercise related to academic achievement. The study showed that early morning exercise with moderate to vigorous intensity optimizes students’ brain function, attentional control skills, and learning attitude [13]. In 2010, the project leader, Dr. John Lately, was invited as a guest lecturer in Korea to introduce his findings and emphasize early morning exercise in the school system. Since then, the early morning exercise program has been implemented at the public schools in the country and was named “0th-period physical activity” or “0th-period physical education class” [14]. Recently, the term “0th-period physical education class (0th-period PE class)” has been generally accepted for autonomous physical activity sessions (i.e., 7:30~8:30 a.m.) before the official class begins [15].

Various studies on “0th-period PE class” have been conducted in Korea, and Jeon et al. [15] found that the participation of middle school students had a statistically significant effect on perceived enjoyment and learning attitudes. Furthermore, an intervention study at the elementary school also concluded that the early morning exercise program significantly improves social skills, autonomy, and stability [16]. Finally, for the high school case, a 3-year long-term morning exercise program positively influenced participants’ fitness test scores, physical self-concept, and academic achievement [17].

Although many studies showed positive results of adolescent morning exercise programs, parents’ involvement and perception are not well documented yet. In general, parental involvement in a child’s education is consistently found to be positively associated with a child’s academic performance [18]. However, Lim et al. [19] pointed out that excessive parental obsession with their children’s academic achievement might harm the expansion of “0th-period PE class”. Furthermore, Jeon et al. [20] stressed that parents’ perceptions should also be further investigated, noting that there will be great differences between middle school students’ perceptions and parents’ perceptions in participation in “0th-period PE class”. In this respect, the Q methodology could be used as one of the prominent ways to reveal other social perspectives that exist on issues or topics related to physical activity [21]. Therefore, considering the parental influence on adolescents, investigating and analyzing parents’ perceptions and demands is of great importance for successfully settling the “0th-period PE class”. The results of this study will contribute to grasping parents’ perspectives in exploring improvements in morning exercise programs and presenting policy directions for future program expansion.

Consequently, the current study examined the parental perception of a morning exercise program for adolescents to provide ideas and agendas for successfully implementing “0th-period PE class” in the country. Two research questions guided the methodology and data analysis. First, using the Q-methodology, how are parents’ perceptions of the “0th-period PE class” categorized? Second, what are the characteristics of each type, and what are the differences, commonalities, and implications?

## 2. Methods

The Q-methodology was developed by William Stephenson in 1953 [22]. It is a suitable technique to recognize the “subjectivity” perceived by humans. Particularly, it is suitable for questions about personal experiences, preferences, values, and beliefs about the research topic [23]. In addition, unlike the R-methodology, the Q-methodology is an operant methodology that represents perceptions in the study subjects’ own thoughts and languages, not in the researcher’s operational definition [24]. For this reason, it is used in different academic fields such as psychology, nursing, and pedagogy [25]. The advantage of the Q method over other forms of discourse analysis is that since everyone responds to the same set of Q statements, researchers can directly compare participants’ responses in a consistent manner. This is usually distinct from other types of qualitative discourse analysis. Hence, it could be an appropriate way to explore how Korean parents perceive their children’s physical activities during the 0th period in South Korea.

### 2.1. Q–Population (Concourse of Statements) and Q-Sample

Q-population, called concourse, is the raw material of the Q-Sample. The Q-population can be constructed using a variety of methods, including focus group interviews, individual in-depth interviews, and literature reviews [26]. Therefore, in this study, Q-population was created through two aspects. First, a focus group interview (FGI) was conducted for a group that included parents with a lot of information about their children’s “0th-period PE class”, physical education (PE) teachers with experience in the class, and professors majoring in sports pedagogy [27]. Second, various documents that can indirectly understand parents’ perceptions of the “0th-period PE class” were analyzed [28]. Through the above method process, 42 Q-populations were finally secured. Next, 42 statements (the “concourse”) were reinterpreted and reclassified. 

In the Q-methodology, the more Q-Samples, the less reliable they tend to be, and if a complex thinking process on the research topic is needed, it is necessary to adopt fewer than 30 [29]. Finally, 25 Q-Samples were confirmed. In addition, Q-sorting was conducted twice for 5 research participants. The test results were found to be r = 0.72, and reliability was secured [30]. The final Q-Sample is presented in Table 1.

### 2.2. P-Sample

The Q-methodology is a method of measuring subjective opinions within an individual. Most importantly, there is no limit to the number of P-Samples because “inter-individual difference in significance” is the core of the study rather than the “inter-individual difference” of the study respondents. This is because the characteristics of the population are not inferred from the characteristics of the P-Sample [31]. The number of P-Samples should be fewer than the number of items in the Q-Sample for statistical reasons [32]. B-middle School in a metropolitan city, South Korea, was selected as a research environment. B-middle School has been running “0th-period PE class” for more than 10 years since 2012. Among them, parents who could provide sufficient data on the “0th-period PE class” were recommended by the teacher in charge of the class. Prior to the start of the study, consent for participation in the study was received from the principal of B school and the physical education teacher. Lastly, 20 parents of students who participated in the “0th-period PE class” for more than one year were selected as P-Samples [22]. The detailed characteristics of P-Samples are as follows in Table 2.

### 2.3. Q-Sorting and Factor Analysis

Q-Sorting is the task of placing Q-Samples in a Q-sorting response table for P-Samples. To this end, a forced sorting method was applied in which 20 P-Samples were classified one by one according to the degree to which they positively or negatively agree with the concept and thoughts of the “0th-period PE class” [33]. The specific Q-Sorting process is as follows. From 1 December 2021 to 31 December 2022, Q-Sorting was conducted for P-Sample. Specifically, due to the recent COVID-19 pandemic, it was conducted through non-face-to-face video conferencing (ZOOM). The Q-Sorting method was sufficiently described to each respondent. In addition, after P-Sample read Q-Sample, the most positive (+4) to most negative (−4) questions were placed at both extremes one by one. Next, the positive, negative, and neutral questions were divided into three groups and placed in the positive (+) to negative (−3). Finally, the contents of the reasons for placing them in the questions of both extremes were written, and interviews were conducted in the background [29]. Regarding this procedure, consent to participate in the study was received before the study began. The distribution frame and score composition of the Q population are shown in Figure 1.

The processing of the collected data was analyzed using the PQ method (VERSION 2.35) program [34]. Factor analysis was conducted through the centroid method, and factor rotation was analyzed through the Varimax rotation method [35]. To determine the ideal number of factors, only factors with an eigenvalue of 1.0 or more were extracted, and the optimal number of types was calculated by entering the number of factors from 7 to 2 [35,36].

## 3. Results

As a result of the parent’s subjectivity in the “0th-period PE class”, a total of 4 types were extracted. The eigenvalue, explanatory variance ratio, and correlation results for each type are as follows. Next, four types of parents’ perceptions of “0th-period PE class” were presented.

### 3.1. Eigenvalues (EVs), Variance and Correlations between Factor (Type)

The EVs for each type appeared as 6.19, 3.42, 3.15, and 3.17, respectively. Furthermore, the explanatory variance was derived as 0.20, 0.15, 0.14, and 0.14, respectively. The total variance was 0.63, resulting in 69% explanatory power for all four factors. All factors met the Kaiser–Guttman criterion [36,37]. Table 3 yields the results.

Table 4 demonstrates the correlation between the four factors. Looking more closely, the correlation between Types 1 and 2 was the highest. On the other hand, Types 2 and 4 had the lowest figures. As most of the overall correlation values were derived low, explanatory power and independence for each type were secured, and it can be seen that all types were clearly distinguished [31].

### 3.2. Type 1 (Factor 1): Urgent Legal and Institutional Settlement of “0th-Period PE Class”: Strengthening Facilities, Programs, Instructors, and Publicity

The results of statements recognized positively or negatively by participants belonging to Type 1 are shown in Table 5. Each Z-score is also presented in Table 5. In this type, the statements that participants positively agreed with are in the following order: Q24, Q23, Q22, and Q21, with Z-scores of 1.80, 1.61, 1.40, and 1.21, respectively. The participants also had the most negative viewpoints on the Q8 statement, with a Z-score of −2.01.

A total of seven participants belonged to Type 1 and showed the largest number of respondents. The P-Sample number and the factor weight were P1 (0.69), P2 (0.74), P8 (0.78), P9 (0.87), P15 (0.88), P17 (0.61), and P18 (0.68). Respondent P15 displayed the highest factor weight, which well represents the point of view of Type 1.

### 3.3. Type 2 (Factor 2): Beachhead for a Vibrant School Life: Improving Interpersonal Relationships, Social Skills, and Adaptability

As shown in Table 5, The statement most agreed upon by the participants in Type 2 is Q4, followed by Q6 and Q5, with a Z-score of 1.82, 1.52, and 1.20, respectively. In addition, the most negative statements is Q9 (Z-score = −1.89), followed by Q8 (Z-score = −1.41) and Q3 (Z-score = −1.20).

Four participants belonged to Type 2. Respondents P3, P10, P16, and P 19 belonged to this type, with factor weights of 0.70, 0.73, 0.90, and 0.81, respectively. The factor weight of P 16 was the highest, indicating well with regard for this type of perspective.

### 3.4. Type 3 (Factor 3): Enhancement of Academic Capability: Increased Concentration, Positive Learning Attitude, and Increased Academic Performance

Table 5 illustrates the positive and negative statements for Type 3 and each Z-score. In Type 3, the most positive Q statement by participants are Q14 (Z-score = 1.92), followed by Q15 (Z-score = 1.65) and Q12 (Z-score = 1.23), whereas the most negative statement is Q3 (Z-score = −1.92).

There are four participants in this type. Participants P4, P5, P11, and P20 belonged to Type 3, and the factor weight was 0.92, 0.70, 0.69, and 0.81, respectively.

### 3.5. Type 4 (Factor 4): Strengthening Physical and Mental Health: Improving Exercise Function, Increasing Interest in PE Classes, and Leading a Healthy Daily Life

The statements (positive and negative) of Type 4 and each Z score are shown in Table 5. The most positively agreed-upon statement is Q10 (Z-score = 2.05), whereas the most negatively agreed statement was Q15 (Z-score = −1.84).

In the last type, Type 4, four respondents were identified. As we have seen from Table 5, participants P6, P12, P13, and P14 were included in this type, and the factor weight was 0.84, 0.89, 0.58 and 0.59, respectively.

### 3.6. Consensus Statements by All Types

Table 6 indicates the commonly agreed-upon statements for all types, namely Q1 and Q2, with Z-scores of 0.78 and 0.60, respectively.

## 4. Discussion

The researchers have discussed the EVs, variance, and correlation between the five types and reviewed the classification results for P-Sample and Q-Sample involved in each type with Z-scores. We have also checked the consensus statements.

In terms of correlation findings regarding each type, since the correlation between all types was low, each type’s independence and explanatory power were secured. A strong correlation between Type 2 and Type 4 was observed, which means that the classification intensity of respondents belonging to Type 2 and Type 4 is the highest. In other words, perceptions between the two types of respondents are different prominently. For example, in the case of Type 2, children’s participation in the early morning exercise program was recognized by the participants belonging to Type 2 as having a positive effect on revitalizing school life. On the other hand, however, they had negative thoughts about physical and mental reinforcement. In the case of Type 4, they were aware of the advantages of improving their children’s exercise skills and increasing interest in PE classes. However, they were not aware of the impact on school life. The discussion based on the results of this study is as follows.

Most of all, respondents from Type 1 showed great awareness of the inclusion of “0-period PE class” into a regular curriculum, not a temporal activity. That is, there is a need for a multifaceted infrastructure in which the “0th-period PE class” can be operated as a regular class. Therefore, expanding facilities, diversifying programs, and placing expert instructors should be considered first to improve the quality of early morning exercise programs. In addition, the respondents were convinced that the “0th-period PE class” helped students develop in various aspects which legitimate promotion expansion of early morning exercise. As a result, this type was titled “urgent legal and institutional settlement of “0th-period PE class”: Strengthening facilities, programs, instructors, and publicity”.

As mentioned earlier, the early morning exercise program has been activated in Korea since 2010. However, lack of attention from the teachers, low incentives for participating teachers and students, and recognition as an extra-curricular activity are regarded as decisive reasons for the institutional failure to settle down. Lee [21] pointed out several issues each school undergoes in the operation of the early morning exercise program. As a result, the researcher suggested optimizing the program for each school environment, strengthening the role of operating teachers, and improving program facilities. Meanwhile, cooperation between PE teachers, general teachers, and students was emphasized as the most crucial factor for the program’s success [19]. For example, Jeon et al. [38] emphasized the theoretical basis for implementing the “0th-period PE class” for middle schools and applied practical measures to encourage the participation of teachers and students. Furthermore, this will be possible through stable administrative and financial support from government and local agencies [21]. Hence, the result that the most significant number of respondents were distributed in Type 1 means the possibility of legal and institutional settlement of the “0th-period PE class”.

Secondly, the Type 2 participants strongly believed that early morning exercise plays a significant role in leading to successful school life. Notably, respondents recognized that the program was helpful in social development, such as cooperation with others, interpersonal skills, and leadership building [39]. Moreover, unlike in regular classes, interaction with friends and teachers frequently occurs, leading to students’ social development and expansion of social networks. These results mean that “0th-period PE class” helps students adapt to overall school life. Lee et al. [40] suggested that the program not only manages stress but also prevents deviations from school for students. Jeon et al. [20] explored Korean middle school students’ subjective perception types and characteristics regarding participation in the “0th-period PE class”. He concluded that the program was recognized as the driving force for students to maintain a stable school and could be an advantage of the morning exercise program from a parent’s point of view.

Based on the discussion above, Type 2 was named “Beachhead for a vibrant school life: Implying interpersonal relationships, social skills, and adaptability”. Meanwhile, participants in Type 2 recognized that “0th-period PE class” had a positive effect on school life but showed a negative perception of improving motor skills and physical fitness such as agility, muscular strength, and endurance.

Thirdly, Type 3 was named “Enhancement of academic capability: Increased concentration, positive learning attitude, and increased academic performance”. In this type, positive perceptions of academic achievement were mainly derived. Interestingly, participants in Type 3 showed a negative perception about the development of social skills through the early morning exercise program.

Recently, scholars in kinesiology have shown scientific evidence of an association between physical activity and academic achievement. Research studies on a positive relationship between physical activity and academic achievement have been discussed [41,42,43], especially for those adolescent populations [4,6]. For example, Hobart [8] demonstrated that the morning exercise program improves the brain’s structure and condition, improving academic performance. Furthermore, there is scientific evidence that physical activity can contribute to developing and activating the hippocampus, which has a significant role in learning and memory in the limbic system [44,45].

Studies conducted in Korea related to these topics produced similar results. For example, participation in the “0th-period PE class” improved high school students’ academic performance than a control group that did not receive the morning exercise program [46]. However, the results also showed that students in the control group had more stress and mental issues. Park and Moon [16] mentioned that the morning exercise program could improve academic performance through increased concentration. Thus, the logical and theoretical connection between morning exercise and academic achievement supports the response of Type 3 participants.

Lastly, Type 4 is titled “Strengthening physical and mental health: Improving exercise function, increasing interest in PE classes, and leading a healthy daily life”. Participants in this category were highly aware of the direct physical effects of participation in sports and exercise. Furthermore, experiencing the morning exercise program contributed to motivation and active participation in PE classes. Based on these results, the healthy daily life of student agency was logically derived. However, respondents of Type 4 showed a negative perception of the impact on academic performance through the early morning exercise program.

The benefits of early morning exercise have been substantially proven by scientific evidence in physiology and medical science [47]. For example, regular morning exercise program participation positively improved adolescents’ motor skills, cardiorespiratory endurance, and muscle strength [16,48]. Furthermore, the sports club activities in the “0th-period PE class” significantly affected elementary school students’ physical self-concept [49]. For this reason, the researchers draw attention to the fact that maintaining healthy daily life is possible by participating in “0th-period PE class” in terms of physical and mental benefits for adolescents [50,51]. Based on the above academic perspective, various perceptions of parents in the results of this study were discussed. A variety of previous studies have already published the effects of early morning exercise. However, this study has the ultimate goal of institutionalizing the “0th-period PE class” based on early morning exercise. Particularly, the positive perception of parents was derived in a different aspect from the students participating [20] in the “0th-period PE class”. As remarked in the introduction, the role of parents is important in the institutional development of public education in South Korea. Thus, this study’s findings can be an academic driving force for “0th-period PE class” to be operated as a regular class in the near future.

## 5. Conclusions

The current study attempted to establish a systematic settlement of the “0th-period PE class” in Korea. The researchers mainly explored the subjective perspectives of parents regarding the morning exercise program of middle schools in Korea using the Q-methodology. The composition of the 42 Q-population resulted in selecting the final 25 Q-Samples. Among the participants, 20 students who participated in “0th-period PE class” for more than one year were selected as P-Sample. The Q-sorting process was performed by the P-Sample. Data collected by Q-sorting were analyzed using the P-Q method program (version 2.35) with centroid factor analysis and varimax rotation. The data analysis categorized the results into four types with a total explanatory variance of 63%. The four types are as follows: urgent legal and institutional settlement of the “0th-period PE class” (Type 1, *n* = 7); beachhead for a vibrant school life (Type 2, *n* = 4); enhancement of academic capability (Type 3, *n* = 4); and strengthening physical and mental health (Type 4, *n* = 4). Moreover, the consensus statements between each type were investigated in Q1 and Q2.

Based on published papers, the morning exercise program is more prevalent and widely spread in the United States, UK, and Japan. Moreover, a legitimate legal basis supports program implementation in some countries, which guarantees adolescents the right to education and a healthy lifestyle [52]. However, the Korean education community has yet to implement legal and institutional procedures to settle the “0th-period PE class”. For this reason, continuous attention of the academic community should be followed [20]. The results of this study produced academic intellectual assets that could be systematically implemented in a “0th-period PE class” in Korea. Future directions include further studies to advance the “0th-period PE class”, as well as better understand the roles and relationships among diverse stakeholders (policymakers, administrators, teachers, students, and parents).

## Figures and Tables

**Figure 1 ijerph-19-07760-f001:**
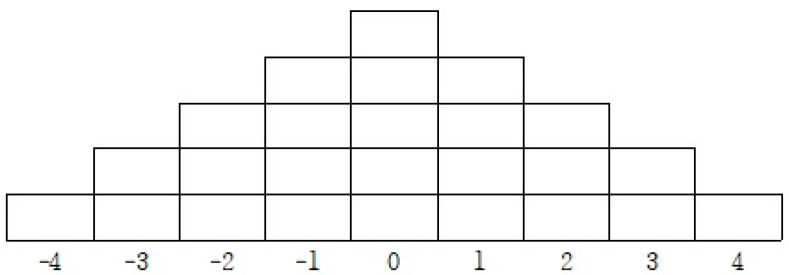
Q sorting response table.

**Table 1 ijerph-19-07760-t001:** Q-Sample.

Q Number	Q Statements
1	Morning exercise can create a sense of goal and impellent.
2	Through participation in morning exercise, it has changed or is becoming a bright personality.
3	Through participation in morning exercise, the relationship with teachers is expanding and improving.
4	Through participation in morning exercise, the relationship with school friends is expanding and improving.
5	Morning exercise helps improve adaptability to overall school life.
6	Morning exercise helps develop social skills such as cooperation and thoughtfulness.
7	Participation in morning exercise is helpful in psychological stability and stress relief.
8	Muscle strength (muscle endurance) and body flexibility improved through morning exercise.
9	Participation in morning exercise helps improve exercise functions such as agility and power.
10	Morning exercise makes physical attractiveness (body, physique, etc.) even better.
11	Participation in morning exercise increases attention and interest in other sports.
12	Morning exercise has increased interest in participating in physical education(PE) class.
13	Morning exercise has given rise to awareness of the importance of future exercise (lifetime PE).
14	Morning exercise increases concentration on various tasks.
15	Through participation in morning exercise, children’s grades improved.
16	Participation in morning exercise reduced the fatigue of children.
17	Participation in morning exercise can develop regular lifestyle habits.
18	Morning exercise induces active and confident daily life.
19	Morning exercise allows youths to develop regular eating habits.
20	It is necessary to change the teacher’s positive perception of “0th-period PE class”.
21	It is necessary to develop infrastructure (facilities, programs) for the operation of the “0th-period PE class”.
22	There is a need for an incentive system for teachers who operate “0th-period PE class”.
23	Research on the effectiveness of “0th-period PE class” is needed.
24	Legal and institutional settlement of “0th-period PE class” is necessary.
25	It is necessary to strengthen the publicity of “0th-period PE class” for parents.

**Table 2 ijerph-19-07760-t002:** Summary of characteristics for P-Sample and factor weight.

Factor (Type)	P-Sample	Age	Child’s Age	Gender	Participation Period (Semester)	Factor Weight
Type 1. (N = 7)	1	55	16	Male	6	0.69
	2	54	15	Male	5	0.74
	8	43	15	Female	5	0.78
	9	52	14	Male	4	0.87
	15	41	15	Female	6	0.88
	17	55	15	Male	6	0.61
	18	45	16	Female	6	0.68
Type 2. (N = 4)	3	44	15	Female	5	0.70
	10	50	16	Male	5	0.73
	16	45	16	Female	5	0.90
	19	51	16	Male	6	0.81
Type 3. (N = 4)	4	49	15	Male	5	0.92
	5	44	16	Female	6	0.70
	11	46	15	Female	5	0.69
	20	44	15	Male	5	0.81
Type 4. (N = 4)	6	56	16	Male	6	0.84
	12	49	14	Female	3	0.89
	13	50	16	Female	5	0.58
	14	58	15	Male	5	0.59
Non-significant sample	7	55	16	Male	6	0.32

**Table 3 ijerph-19-07760-t003:** Eigenvalue (EVs) and variance between types.

	Type 1	Type 2	Type 3	Type 4
**Eigenvalue (EVs)**	6.19	3.42	3.15	3.17
**% of explanatory variance**	0.20	0.15	0.14	0.14
**Total variance**	0.20	0.35	0.49	0.63

**Table 4 ijerph-19-07760-t004:** Correlations between types.

	Type 1	Type 2	Type 3	Type 4
**Type 1**	1.000			
**Type 2**	0.16	1.000		
**Type 3**	0.08	0.40	1.000	
**Type 4**	0.12	0.05	0.13	1.000

**Table 5 ijerph-19-07760-t005:** Statements with a Z score of ±1.00 or higher for each type and Z score results.

		Q Statement	Z Score
**Type 1**	**Positive**	Q 24	1.80
		Q 23	1.61
		Q 22	1.40
		Q 21	1.21
	**Negative**	Q 8	−2.01
		Q 9	−1.69
		Q 11	−1.35
**Type 2**	**Positive**	Q 4	1.82
		Q 6	1.52
		Q 5	1.20
	**Negative**	Q 9	−1.89
		Q 8	−1.41
		Q 3	−1.20
**Type 3**	**Positive**	Q 14	1.92
		Q 15	1.65
		Q 12	1.23
	**Negative**	Q 3	−1.92
		Q 4	−1.50
		Q 6	−1.19
**Type 4**	**Positive**	Q 10	1.82
		Q 8	1.41
		Q 12	1.17
	**Negative**	Q 15	−1.84
		Q 14	−1.30

**Table 6 ijerph-19-07760-t006:** Consensus statements.

Q Statement	Z Score
Q 1. Morning exercise can create a sense of goal and impellent.	0.78
Q 2. Through participation in morning exercise, it has changed or is becoming a bright personality.	0.60

## Data Availability

The data presented in this study are available upon request from the corresponding authors.

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
