# Peer review of "A Study on the Subjectivity of Parents Regarding “0th-Period Physical Education Class” of Middle Schools in Korea Using Q-Methodology"

_ijerph, 2022, doi:10.3390/ijerph19137760_

Round 1

Reviewer 1 Report

Overall:  A very well-written paper on the study of parental perspectives of pre-school physical activities in Korean children. 

Abstract:  Abstract is very clear and concise and covers the relevant information regarding methodology, findings, and general conclusions.

Introduction:  The introduction does a very good job of creating the context for the study and highlighting previous research in morning exercise, the effects of exercise on both health and cognition, and the importance of involving parental engagement. As the abstract  mentioned, the lack of studies involving parents helps to highlight the need for this study. 

Methods:  The first section highlights Q-Methodology and provides an overall view of the method and its importance.  It would be helpful to provide additional details regarding the Q-methodology and what it is compared to other methods, specifics as to what it entails, and how it’s typically used.  While a percentage of the readership may be familiar with this method, it would be helpful for those that aren’t.

Additional details regarding the meaning of a Q-population and P samples are very helpful so more information about the methodology itself would also prove beneficial.

Results:  The results are well-described and easy to follow with appropriate definitions and explanations of findings.

Discussion:  The discussion does a great job of covering the importance of the study in the context of Korean culture and historical use of exercise in early morning activities prior to traditional school.  The discussion and conclusion sections are missing elements related to discussion the limitations of the study. 

Author Response

Dear Reviewer 1;

First of all, I am deeply grateful for your thoughtful comments.

I really appreciate regarding reviewers' detailed comments. Also, I would like to express my sincere gratitude for your evaluation.

I fully understand what you are referring to in the “comments about Method & Discussion sector” and I/we’ve sincerely tried to revise the method & discussion part according to the reviewer's point of view. Moreover, I marked the sentence modified.

Additionally, there seems to be a misunderstanding in the parts mentioned by the reviewer and it seems that the misunderstanding occurred because there a deep understanding of the Q methodology was not accompanied. As mentioned above, in general, the sampling method in the Q method is conducted based on previous studies. Please refer to the following documents (provided and revised in this paper). Particularly, these are pioneer (Stephenson, W) and developers (Baker, R. M & Brown, S. R.) of Q methodology, and their logic has already been verified in a wide variety of disciplines. We also conducted this paper based on this logic. (All theoretical logic was cited in our manuscript.)

 In summary, we tried to supplement the discussion part additionally according to your valuable point. However, I ask for your generous understanding of the part related to the Q methodology I mentioned.

Accordingly, I/we have done our best to revise our manuscript, and thank you very much.

Best Regards,

Reviewer 2 Report

Thank you for the opportunity to review this valuable work. The contents that must be reviewed in your manuscript are as follows.

1. Introduction section

- I think it is necessary to conduct a survey on the perception of participating students in order to find improvements in the morning exercise program. Please describe in detail why this study is necessary.

Also, why is it aimed at parents of middle school students?

2. Method section 

- Has consent been obtained for the information of study subjects used in the study? Please describe what procedure was used.

- Was the survey done by one institution? Since the survey was conducted non-face-to-face, it could be conducted on many study subjects. However, the number of study subjects is very small. This may make it difficult to reliability the results of this study. It seems that it should be conducted in various regions and various institutions. 

3. Discussion section

Please describe the limitations of this study.

Author Response

Dear Reviewer 2;

First of all, I am deeply grateful for your thoughtful comments.

I really appreciate your detailed comments. Moreover, I would like to express my sincere gratitude for your evaluation and advice.

First, you mentioned:

  1. Introduction section

- I think it is necessary to conduct a survey on the perception of participating students in order to find improvements in the morning exercise program. Please describe in detail why this study is necessary.

Also, why is it aimed at parents of middle school students?

I fully understand the comment you mentioned. The research on students' perceptions you mentioned was conducted by me and my colleagues before. Please refer to the link.

https:// doi.org/10.3390/su132112081

In the previous study, the study was started in that parents' perceptions would be slightly different. Of course, there was no prior study. From this point of view, it was additionally described why parents’ perceptions should be explored in the introduction sector. Thank you.

Second, you mentioned:

  1. Method section 

- Has consent been obtained for the information of study subjects used in the study? Please describe what procedure was used.

- Was the survey done by one institution? Since the survey was conducted non-face-to-face, it could be conducted on many study subjects. However, the number of study subjects is very small. This may make it difficult to reliability the results of this study. It seems that it should be conducted in various regions and various institutions. 

I fully understand what you are referring to in the “comments about the Method sector” and I/we’ve sincerely tried to revise the method part according to the reviewer's point of view. Moreover, I marked the sentence modified.

Additionally, there seems to be a misunderstanding in the parts mentioned by the reviewer and it seems that the misunderstanding occurred because there a deep understanding of the Q methodology was not accompanied. As mentioned above, in general, the sampling method in the Q method is conducted based on previous studies. Please refer to the following documents (provided and revised in this paper). Particularly, these are pioneer (Stephenson, W) and developers (Baker, R. M & Brown, S. R.) of Q methodology, and their logic has already been verified in a wide variety of disciplines. We also conducted this paper based on this logic. (All theoretical logic was cited in our manuscript.)

 In summary, we tried to supplement the discussion part additionally according to your valuable point. However, I ask for your generous understanding of the part related to the Q methodology I mentioned.

Third, you mentioned:

  1. Discussion section

Please describe the limitations of this study.

I am deeply grateful for your thoughtful comments. The part about the conclusion has been appropriately modified.

Lastly, I/we have done our best to revise our manuscript, and thank you very much.

Best Regards,

Reviewer 3 Report

It is not clear what new things this paper has discovered. There is a statement at the end of the paper that there were similar studies in Japan and the United States, but it should be mentioned in the first half. After carefully discussing previous research in the first half, you should point out the lack of previous research and clearly state the hypothesis that this paper tests. The discussion section of the paper should clearly discuss what the outcome was for the hypothesis. Also, please enrich the discussion about the meaning of this paper in academia and society.

Author Response

Dear Reviewer 3;

First of all, I am deeply grateful for your thoughtful comments.

I really appreciate regarding your detailed comments. Moreover, I would like to express my sincere gratitude for your evaluation and advice.

First, you mentioned:

It is not clear what new things this paper has discovered. There is a statement at the end of the paper that there were similar studies in Japan and the United States, but it should be mentioned in the first half. After carefully discussing previous research in the first half, you should point out the lack of previous research and clearly state the hypothesis that this paper tests. The discussion section of the paper should clearly discuss what the outcome was for the hypothesis. Also, please enrich the discussion about the meaning of this paper in academia and society.

I fully understand the comment you mentioned. This study does not simply verify the effectiveness of morning exercise. The overseas study you mentioned refers to academic knowledge of the broad concept of morning physical activity. However, this study aims to produce academic knowledge that "0 period physical education class" can be operated at an institutional level in Korea. Based on these points, the discussion was re-stated and supplemented more abundantly.

I am deeply grateful for your thoughtful comments. The part about the discussion and conclusion has been appropriately modified.

Round 2

Reviewer 3 Report

The author has made the corrections carefully.